# Bridging Audio-Visual Semantics with Language-Guided Synthesis

## Abstract

One of the underlying assumptions behind audio-visual learning models is that the two modalities convey overlapping information. However, this assumption is widely violated in practice, which results in degraded performance. To address this problem, we propose to replace mismatched audio-visual signals using cross-modal generative models. Our approach uses language-based supervision to perform this generation. We show that data synthetically generated through this process is well-suited for a variety of audio-visual representation learning methods. The features that we learn this way outperform those trained solely on real data for a range of downstream tasks, including audio classification, audio-visual retrieval, and visual sound localization.

## 1 Introduction

Audio-visual representation learning remains a challenging open problem, in large part due to the scarcity of high-quality paired multimodal data. While there are vast amounts of paired audio-visual data available on the internet, the images and audio often convey very different information. Sound sources, for example, are often not visible, due to being distant, occluded, or outside the camera's narrow field of view. While recent feature learning methods are designed to tolerate *some* degree of mismatch between modalities, these misaligned examples collectively degrade the quality of the learned representations.

In this paper, we use generative models to create synthetic *audio-visual* data for audio representation learning, taking inspiration from work that generates training data for visual learning (Tian et al., 2023b; Azizi et al., 2023; He et al., 2023; Tian et al., 2023a). Given a natural video dataset, we replace images that do not match the video's corresponding audio through cross-modal audio-to-image prediction. Since these generated images convey the same scene structures as the audio, the resulting dataset is more suitable for multimodal representation learning methods. We also extend this approach by generating synthetic audio conditioned on visual inputs to substitute low-quality audio signals.

To perform cross-modal generation, we use text-conditioned generation models, treating language as a bridge between the modalities. Specifically, we generate a caption for a given example and then pass it into a cross-modal prediction model, allowing us to leverage large-scale pretrained generation and captioning models.

There are several challenges when creating a multimodal audio-visual synthetic dataset. In contrast to other synthetic dataset generation work (Tian et al., 2023b;a), the captions here are rooted in one of the two modalities rather than being created from scratch, constraining their content. Moreover, the captions generated by an off-the-shelf vision-language model will not necessarily reference the sound sources that are present in the audio, leading to mismatched images and audio. Finally, training solely with synthetic data fails to take advantage of the subset of real examples in the dataset that *are* well-aligned and may limit the ability to process real examples, which can be subtly out-of-distribution. We address these issues by rewriting text prompts to convey sound source information and by mixing real and synthetic data through a filtering process.

We evaluate our approach by generating a synthetic version of the VGGSound dataset (Chen et al., 2020; 2021). We assess the quality of the generated data using various representation learning methods, including audio-visual contrastive learning (Guzhov et al., 2021; Wu* et al., 2023; Luo et al.,

2023) and masked autoencoders (He et al., 2021; Georgescu et al., 2023; Gong et al., 2023). We observe that the resulting representations outperform those trained on the original dataset in downstream tasks on popular datasets such as ESC50 (Piczak, 2015), FSD50k (Fonseca et al., 2022), Urban8k (Salamon et al., 2014), and VGGSound (Chen et al., 2020). Through human ratings and automated evaluation metrics, we also verify that the generated audio-visual pairs are more semantically aligned.

These experiments suggest:

- Simply replacing all images in VGGSound (Chen et al., 2020) with synthetic images is helpful for audio representation learning, outperforming models trained with real images on audio recognition tasks.
- Models that use a mix of real and synthetic data outperform those trained on real or synthetic data alone.
- The representation learned from our generated data performs well on a variety of downstream tasks, including audio classification, cross-modal retrieval, and audio-visual sound localization.

## 2 RELATED WORK

**Audio-visual representation learning.** A variety of early methods learned audio-visual representations from unlabeled data through cross-modal prediction (Ngiam et al., 2011; De Sa, 1993; Owens et al., 2016b;a; Asano et al., 2020). One successful approach is to use contrastive learning (Arandjelovic & Zisserman, 2017; Luo et al., 2023; Morgado et al., 2021; Ma et al., 2021; Sarkar & Etemad, 2022; Owens & Efros, 2018) to bring the embeddings of paired audio-visual samples together, while pushing apart non-paired samples. These contrastive methods have also been applied directly to sound localization (Senocak et al., 2018a; Arandjelovic & Zisserman, 2018; Owens & Efros, 2018; Mo & Morgado, 2022; Sun et al., 2023; Senocak et al., 2024; Chen et al., 2021), where they select image regions that correlate strongly with sounding objects. A recent line of work applies masked autoencoders to learn audio-visual representations. Early methods such as AV-MAE (Georgescu et al., 2023), TVLT (Tang et al., 2022), and MAViL (Huang et al., 2023) predict masked visual and audio tokens using a joint decoder. CAV-MAE (Gong et al., 2023) improves performance by combining contrastive learning with masked modeling. CAV-MAE Sync (Araujo et al., 2025) further improves CAV-MAE by aligning audio and iamge features on a fine-grained space. Zhu et al. (Zhu & Duan, 2024) extend these methods by adding text using an autoregressive captioner. Su et al. (Su et al., 2024) propose using image tokens to predict masked audio tokens with an encoder-decoder architecture for both representation learning and generation. These approaches implicitly assume that audio-visual pairs convey the same scene structure, and they degrade in performance if one modality cannot be predicted from the other. Our method is complementary: we apply these self-supervised learning approaches to *synthetic* data that is specifically designed to provide a useful multimodal learning signal.

**Language for audio-visual representation learning.** This approach was popularized by CLIP (Radford et al., 2021), which learns joint embeddings of language and vision from large-scale paired data. Subsequent models, such as CLAP (Wu* et al., 2023; Chen et al., 2022), extend this paradigm to audio–text representation learning. Other works, including VatLM (Zhu et al., 2024) and AudioCLIP (Guzhov et al., 2021), further generalize to tri-modal embeddings across text, audio, and vision. Our work is also related to approaches such as ImageBind (Girdhar et al., 2023), which connect multiple modalities by aligning them to frozen vision–language embeddings. Ex-MCR (Zhang et al., 2024) aligns audio and image features with pretrained CLIP and CLAP features, and LG-CAV-MAE (Ishikawa et al., 2025) employs captioning to associate audio and visual signals with text, but does not address the underlying cross-modal mismatches. In contrast, we leverage language not only as an alignment space but as an intermediary to generate semantically consistent counterparts, enabling the repair of noisy or corrupted audio–visual pairs.

**Representation learning using synthetic data.** To address data scarcity and quality issues, synthetic data has been used for representation learning (Azizi et al., 2023; Tian et al., 2023b;a; He et al., 2023). Recent advances in large language models enable text rewriting for multimodal data augmentation (Fan et al., 2023; Zhu & Duan, 2024). In computer vision, synthetic image datasets can be generated through diffusion models (Tian et al., 2023b; Azizi et al., 2023; He et al., 2023;

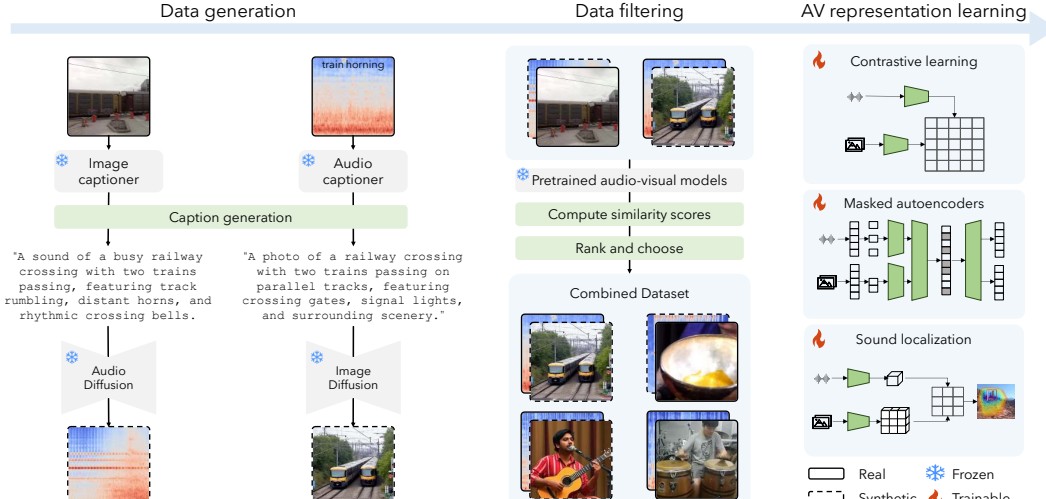

Figure 1: **Generating and learning from synthetic audio-visual data.** First, we generate a dataset that combines real and synthetic data. We create candidate synthetic audio-visual pairs by captioning the audio, refining these captions using a language model, then generating an image using a text-to-image model (and vice versa for audio generation from images). We then construct the dataset by mixing real audio-visual pairs that have high semantic similarity, as measured by cosine similarity of a joint embedding, with our synthetic data. Second, we use this combined dataset for representation learning by applying off-the-shelf audio-visual feature learning methods.

Tian et al., 2023a). Synthetic data has also been used to improve sound localization in audio-visual tasks by leveraging alignment between modalities (Senocak et al., 2024; 2023). The most related to our work is AVAgent (Mo & Song, 2024), which employs LLMs to edit audio signals for better alignment. In contrast, our method performs full generative repair, allowing us to reconstruct audio-visual pairs even when the original samples are entirely corrupted, rather than merely refining partially mismatched inputs. Our language caption can also be used as a signal during the pretraining process, which further boosts the performance (Ishikawa et al., 2025).

One of the primary challenges with synthetic data is its limited distribution (Gerstgrasser et al., 2024; Seddik et al., 2024; Feng et al., 2024). Efforts to address this have focused on improving the diversity of generated datasets via techniques like prompt engineering (Tian et al., 2023a). A recent line of work replaces generative models with retrieval to create synthetic datasets (Geng et al., 2024). This approach is complementary to ours and could potentially be incorporated into our pipeline in the future.

## 3 METHOD

We use synthetic data to create a dataset that is well-suited for self-supervised audio-visual representation learning. As shown in fig. 1, we build this dataset by combining real and synthetic data. We generate the synthetic data using cross-modal prediction, then apply a filtering procedure to select real and synthetic examples with a high degree of alignment. We then use our combined dataset to train audio-visual representation learning methods.

### 3.1 GENERATING SYNTHETIC DATA

We first use cross-modal prediction to generate *candidate* synthetic data from an existing audio-visual dataset (in our experiments, we use VGGSound (Chen et al., 2020)). For each audio-visual pair in the dataset, we caption both modalities, then use text-to-image and text-to-audio models to produce candidate synthetic data.

A key challenge in synthetic data generation is the issue of narrow distribution. Simple prompts such as "A photo of a car" often lead to a limited range of images that look similar, even when the guidance rate is low. This lack of diversity can significantly hinder pretraining performance, especially as the dataset size grows. Previous works like SynCLR (Azizi et al., 2023) address this

issue by using context learning-based methods to introduce diversity into the generated captions, typically employing large banks of templates and environment lists that make captions more vivid. However, these methods generally focus on two modalities, so any changes to the prompt still ensure the generated images align with the text. In our case, the caption is only an intermediate step, so too much modification can lead to hallucination by the language model (Tian et al., 2023a; Xu et al., 2024) and produce captions that no longer align with the audio, harming downstream tasks.

Therefore, instead of inserting too many additional entities (as in Tian et al. (2023a)), we balance caption vividness and correctness by altering only the environment and perspective, while leaving the main object largely unchanged. Take audio-to-image generation as an example: in the caption synthesis step, we ask the model, "What are the sounding objects in the audio?" After getting the answer (the captioner is itself a language model), we then ask, "Giving the sounding objects, what is the reasonable environment in which this sounding event may happen?" We finally have the language model combine the environment and the objects into a prompt for image generation, such as: "Could you please detailedly describe a scene by placing the sounding objects into that environment, specify the foreground and background." In addition, we use a list of negative prompts, such as 'non-photometric', 'poor light condition' and 'distortion', to reduce image defects and further enhance image quality.

## 3.2 Combining Synthetic and Real Data

Due to the distribution gap between real and synthetic data, we aim to minimize distribution shift at test time. Since vision representations can already be effectively learned from abundant vision–text data, our focus is on audio downstream tasks. Specifically, pairing real audio with synthetic images is effective because it ensures that the audio modality—used during testing—comes from the same distribution as in training, while still benefiting from the complementary synthetic counterpart.

However, because real data inherently has some degree of correct alignment, we do not want to discard it entirely; instead, we retain high-quality pairs in our dataset. Moreover, a fully synthetic version of one modality may cause out-of-distribution issues, as the real signal's learned features can become overly similar to those of its synthetic counterpart. Including real data thus serves as a form of augmentation, helping the model generalize better.

To ensure the dataset maintains high-quality alignment, we develop a filtering pipeline that automatically selects the best-aligned real audio-visual pairs. We measure alignment using the CLIP score (Hessel et al., 2022), extracting features through large pre-trained models such as ImageBind (Girdhar et al., 2023) and computing cosine similarities in a shared embedding space. One challenge is that the distribution used to pre-train these models may differ from our target dataset, making a fixed threshold unreliable. To address this, we sort all audio-visual pairs by cosine similarity and retain the top-ranked pairs based on a predefined ratio. Another issue is that some real samples may be noisy or corrupted; to handle this, we remove the 1% of pairs with the lowest alignment scores and replace 5% of the remaining low-scoring audio with synthetic samples from the image.

## 4 Experiments

We first conduct a human study and an automatic evaluation to assess the dataset's quality. We then further evaluate the generated data by pre-training various self-supervised representation learning methods and testing their performance on a range of downstream tasks.

We experimentally evaluate our dataset in two ways. First, we evaluate our approach's ability to generate high quality synthetic data. Second, we evaluate our dataset's suitability for audio-visual representation learning.

### 4.1 Implementation Details

**Data generation pipeline.** We use individual video frames for the visual signal because this is the most common setting for audio-visual representation learning (Huang et al., 2023), and generating full video is a challenging open problem. This choice enables the use of standard representation learning and image generation methods for our evaluation.

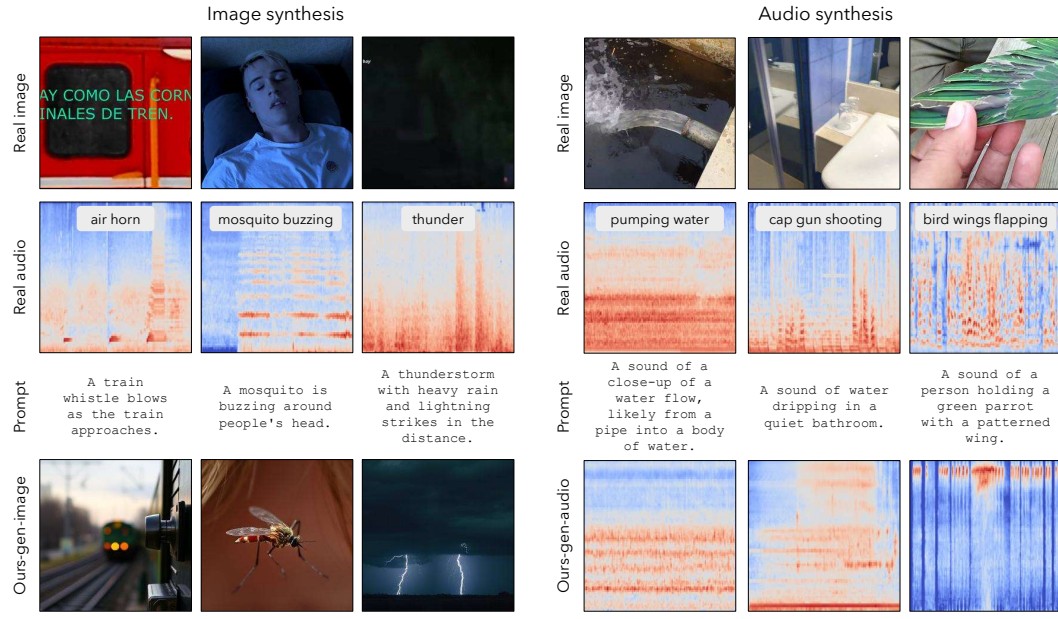

Figure 2: **Qualitative examples from our dataset.** We selected several samples from the generated dataset, showing audio-to-image synthesis (top) and image-to-audio synthesis (bottom). We show the corresponding real data from VGGSound (Chen et al., 2020), with the ground truth text label shown on the mel-spectrogram. We show the prompt used for generation, followed by the generated result.

For audio-to-image generation, we apply SALMONN (Tang et al., 2024; Sun et al., 2024) to caption the audio and then use Llama-7B (Touvron et al., 2023) for caption rewriting. We generate images from these captions using FLUX.1-schnell (Lipman et al., 2023), a latent flow matching model distilled for few-shot generation. For image-to-audio generation, we employ LLAVA-7B (Liu et al., 2023) to generate captions from the input images, and then convert text to audio using Stable-Audio-Open-1.0 (Evans et al., 2024). Unless otherwise specified, we use default hyperparameters for all off-the-shelf models.

**Dataset construction.** To evaluate our data generation pipeline, we train representation learning methods on both real and synthetic versions of a dataset and compare their performance on downstream tasks. We observe that training exclusively with synthetic data yields poor results across many metrics (table 3), likely due to distribution differences between generated and real data.

To mitigate this issue, we use real audio as the primary modality and replace the images with synthetic counterparts during representation learning. We construct two main versions of dataset. One dataset consists entirely of real audio paired with synthetic images, ensuring that the learned audio representations are grounded in real sound. To study the effect of mixing modalities, we also create another dataset in which 5% of the worst real audio is substituted with synthetic audio and 5% most paired real images is maintained. Finally, we perform an ablation study to systematically investigate how different real-to-synthetic ratios influence performance.

**Representation learning methods.** For representation learning methods, we select several widely used techniques, including contrastive learning models CAVP (Luo et al., 2023), which learns audio-visual embeddings, CLAP (Wu* et al., 2023), which learns audio-language embeddings, and AudioClip (Guzhov et al., 2021), which learns joint embeddings between all three modalities. We also evaluated a method based on masked autoencoders, CAV-MAE (Gong et al., 2023). We apply the learned representation models to audio related downstream tasks, such as sound events classification, along with audio-visual applications like audio-visual retrieval and visual sound localization. We hypothesize that if the audio and visual signals in a dataset convey the same semantic information, then the representations that we learn will perform better on these downstream tasks.

**Datasets and training details.** Following prior work (Gong et al., 2023; Mo & Song, 2024), we evaluate our pipeline on the VGG-SS Chen et al. (2021), a 144K subset of the full vggsound dataset (Chen et al., 2020). We follow AudioCLIP's preprocessing protocol (Guzhov et al., 2021),

Table 1: **Perceptual study.** We conduct a human perceptual study to evaluate the quality and alignment of our synthesized data.

| Task | Prefer real | Prefer synth. | Both good | Both bad |
|---|---|---|---|---|
| Audio → Image | 18.9 | **46.7** | 25.5 | 8.9 |
| Image → Audio | 21.1 | **55.6** | 15.5 | 7.8 |

Table 2: **Automatic dataset evaluation.** We measure the similarity between the visual and audio signals by captioning them both, their similarity using Sentence-BERT (Reimers & Gurevych, 2019) embeddings. The 'Real' and 'Syn.' mean whether this signal is real or generated.

| Visual | Audio | Similarity (Reimers & Gurevych, 2019) |
|---|---|---|
| Real | Real | 38.7 |
| Real | Syn. | 41.1 |
| Syn. | Real | **52.1** |

extracting the middle frame of each video as the image input and resampling the audio to 16kHz. Because the middle frame may not always contain the sounding object, we also consider a stronger "best frame" baseline. In this baseline, we sample 10 frames at uniform intervals from each video, then use a pre-trained audio-visual model (Girdhar et al., 2023) to pick the frame most strongly correlated with the audio.

We use linear probing experiments to assess the learned representations, where we test the [CLS] token's representation on ESC-50 (Piczak, 2015), FSD-50k (Fonseca et al., 2022), Urban-8k (Salamon et al., 2014), and VGGSound test set (Chen et al., 2020). We also evaluate visual sound localization by using EZ-VSL (Mo & Morgado, 2022) as the localization backbone, training on different VGG-SS variants and testing on Flickr-SoundNet (Senocak et al., 2018b; 2020) and VGG-SS's test set (Chen et al., 2021). All models are trained on 4 NVIDIA A40 GPUs, with 30 epochs for pretraining and 10 epochs for downstream tasks. Unless otherwise specified, all models are trained from scratch. We use the default hyperparameter configurations provided by LAION-CLAP (Wu* et al., 2023) and CAV-MAE (Gong et al., 2023).

## 4.2 EVALUATION OF THE SYNTHETIC DATASET

First, we use a human perceptual study and automated metrics to assess how semantically aligned the visual and audio signals are in our synthetic dataset. Specifically, we investigate whether the audio and visual signals in our generated data convey the same scene content more consistently than their counterparts in the original, real dataset.

**Perceptual study.** We randomly selected 500 image-audio pairs from VGGSound (Chen et al., 2020). Participants were presented with a reference set consisting of a real image-audio pair, along with a synthesized version (in which either the image or audio was generated). They were asked to decide which option was more aligned with the reference, or to indicate if both were equally good or poor. Each participant evaluated 10 image-generation samples and 10 audio-generation samples, all selected randomly from the full set.

As shown in table 1, participants generally preferred synthetic data over real data. In over 46.7% of the cases, they found the generated images more representative of the reference audio, and in 55.6% of the cases, they found the generated audio more representative of the reference image. Both differences were statistically significant ($p \leq 0.0003$ under a two-sided $t$-test).

Fig. 2 shows example image-audio pairs from VGGSound, along with their corresponding synthetic versions. We also include the captions used for the generation process. The real images often contain distortion, are not photorealistic, or have poor lighting. For audio, the real samples sometimes include background noise (e.g., in the "pumping water" example) or distant speech sounds (e.g., in the "cap gun shooting" example). In contrast, the synthetic audio is typically clearer and more single-source.

In both image and audio cases, the generated content is more semantically aligned with its paired counterpart. This is partly due to the generated captions being more detailed than the original labels and more consistently capturing relevant elements from the audio or visual input. For instance, the thunder example's label omits rain sounds. By fully capturing the sounding objects, the generated

Table 3: **Linear probing results on audio classification.** Audio classification accuracy using different representation learning methods pretrained on different variations of real and synthetic versions of VGG-SS (Chen et al., 2021; 2020). In the data type section, $\mathcal{A}$, $\mathcal{V}$, $\mathcal{T}$ mean the data type for audio, visual, and text, where 'Real' indicates that this modality in this variation is composed of real data, 'Syn.' denotes that the modality is entirely generated, and 'Comb.' represents a mixture of both real and synthetic data. For the image modality, 'MF' means middle frame and 'BF' indicates it is using the selected best frame. For the text modality, 'Label' signifies the use of original text labels as model input, while 'Cap.' refers to using the audio's caption as input.

| Method | Data Type | | | ESC-50 (Piczak, 2015) | FSD-50k (Fonseca et al., 2022) | Urban-8k (Salamon et al., 2014) |
|---|---|---|---|---|---|---|
| | $\mathcal{A}$ | $\mathcal{V}$ | $\mathcal{T}$ | | | |
| CAV-MAE (Gong et al., 2023) | Real | Real$_{MF}$ | - | 77.8 | 35.1 | 82.5 |
| | Real | Real$_{BF}$ | - | 83.9 | 36.2 | 83.1 |
| | Real | Syn. | - | 86.5 | **38.3** | 83.6 |
| | Comb. | Comb. | - | **87.0** | 36.2 | **84.0** |
| CAVP (Luo et al., 2023) | Syn. | Real$_{MF}$ | - | 33.1 | 24.8 | 56.9 |
| | Real | Real$_{MF}$ | - | 72.8 | 40.1 | 74.3 |
| | Real | Real$_{BF}$ | - | 76.5 | 40.9 | 76.3 |
| | Real | Syn. | - | 82.0 | 45.7 | 81.4 |
| | Comb. | Comb. | - | **83.8** | **46.2** | **82.8** |
| CLAP (Wu* et al., 2023) | Real | - | Label | 83.3 | 46.1 | **81.5** |
| | Real | - | Cap. | 84.3 | **46.3** | 80.1 |
| AudioCLIP (Guzhov et al., 2021) | Real | Real$_{MF}$ | Label | 83.0 | 45.7 | 77.5 |
| | Real | Real$_{MF}$ | Cap. | 85.8 | 46.2 | 78.9 |
| | Real | Real$_{BF}$ | Cap. | 85.5 | 46.3 | 80.1 |
| | Real | Syn. | Cap. | 86.5 | 46.8 | **82.8** |
| | Comb. | Comb. | Cap. | **87.0** | **47.1** | 82.0 |

Table 4: **Diversity evaluation using Vendi score.** We compute Vendi scores on 3,000 randomly sampled generated images. Prompt refinement substantially improves diversity.

| Generation method | Vendi score |
|---|---|
| Without prompt refinement | 15.9 |
| With prompt refinement (ours) | **21.9** |

images become better aligned with the audio. Similarly, the audio generation pipeline only includes objects depicted in the scene, avoiding out-of-scene noises.

**Automatic evaluation.** Inspired by Girdhar et al. (2023), we evaluate the semantic similarity between images and audio using language, as it provides a unified domain containing semantically rich information. For this evaluation, we apply SALMONN (Tang et al., 2024) and LLAVA (Liu et al., 2023) to caption both the real and synthetic images and audio. We then compare their embeddings using Sentence-BERT (Reimers & Gurevych, 2019) via cosine similarity. As shown in table 2, the similarity scores increase for both synthetic images and synthetic audio.

**Diversity and control analysis.** To further assess the diversity of the generated images, we compute the Vendi score (Friedman & Dieng, 2023) on a randomly selected subset of 3,000 generated samples. As shown in Table 4 Images produced without our prompt refinement achieve a Vendi score of 15.9, whereas those generated with our prompt template reach 21.9, indicating substantially higher diversity. Qualitative comparisons can be found in fig. 2.

### 4.3 REPRESENTATION LEARNING EVALUATION

Next, we train models on our synthesized dataset and evaluate the learned representations on various downstream tasks.

**Classification.** We present linear probing results for several pretraining methods (Luo et al., 2023; Gong et al., 2023; Guzhov et al., 2021) on popular audio classification benchmarks in table 3. Additionally, we include audio-visual classification results in table 5. Our findings show a notable performance boost across all classification tasks when models are pretrained on our synthetic dataset, compared to those trained solely on real data—even when using the *best-aligned* frame from each video. This improvement suggests that synthetic data can significantly enhance downstream performance on a variety of tasks. Furthermore, we observe that the synthetic data benefits a wide range

Table 5: **Linear probing results on audio-visual classification.** Audio-visual classification accuracy with CAV-MAE (Gong et al., 2023). Results are reported on the test split of VGGSound (Chen et al., 2020).

| Method | Data Type | | VGGSound (Chen et al., 2020) |
|---|---|---|---|
| | $\mathcal{A}$ | $\mathcal{V}$ | |
| | Real | $Real_{MF}$ | 50.9 |
| CAV-MAE (Gong et al., 2023) | Real | $Real_{BF}$ | 51.2 |
| | Real | Syn. | 51.5 |
| | Comb. | Comb. | **52.7** |

Table 6: **AudioSet pretraining comparison.** CAV-MAE pretrained on real vs. synthetic AudioSet (1.7M pairs), evaluated on VGGSound.

| Pretraining data | VGGSound audio cls. | VGGSound audio–visual cls. |
|---|---|---|
| Real AudioSet | 58.2 | 61.9 |
| Synthetic AudioSet (1.7M) | **58.9** | **62.5** |

of algorithms, including contrastive methods like CAVP (Luo et al., 2023) and reconstruction-based methods like CAV-MAE (Gong et al., 2023).

Notably, leveraging triple-modality aligned data (audio, visual, and text) yields superior audio classification results. Initially, AudioCLIP and CLAP perform similarly—or even worse—when trained on real data, likely because uncorrelated pairs hinder the learning process. However, when trained on our synthesized data, AudioCLIP significantly outperforms CLAP, highlighting the advantages of using synthesized multimodal data for representation learning.

**AudioSet experiments.** To further assess scalability and robustness, we apply our pipeline to AudioSet (Gemmeke et al., 2017). Due to dead links in the original dataset, our synthetic reconstruction contains 1.7M image–audio pairs. We train a CAV-MAE model (Gong et al., 2023) on this synthetic AudioSet and compare it to the same architecture trained on the full real AudioSet. For a fair comparison, both models are fine-tuned on VGGSound with identical downstream architectures and hyperparameters. As shown in Table 6, on VGGSound audio classification, the model pretrained on synthetic AudioSet achieves an accuracy of 58.9, compared to 58.2 for the model pretrained on real AudioSet. For audio-visual classification, our synthetic-pretrained model reaches 62.5, while the real-data baseline achieves 61.9. These results indicate that, even with fewer training pairs, synthetic AudioSet can yield representations that slightly outperform those learned from the full real dataset.

**Cross-modal retrieval.** We evaluated the retrieval performance of AudioCLIP (Guzhov et al., 2021) and CAV-MAE Gong et al. (2023) trained on synthetic and real images (see table 7) using the VGGSound test set. Besides the full test set, we also constructed a "clean" subset because, like the training set, the original test set contains numerous misaligned samples. We created this subset by calculating the CLIP score (Hessel et al., 2022) between each VGGSound label and its corresponding image, removing samples with scores below 25. Results on both the original test set and this cleaned subset are reported. We also evaluated category-level retrieval performance, where a match is counted if the retrieved audio or image belongs to the same category as the reference.

In zero-shot retrieval tasks, models trained on synthetic data underperformed compared to those trained only on real data, especially when tested on the full, unfiltered dataset. Beyond data quality issues, this disparity may stem from the distribution gap between synthetic and real images, making it difficult for the visual branch trained on synthetic data to map real images into the same latent space as synthetic ones. Consequently, the embeddings become misaligned with the audio, leading to lower retrieval scores. Notably, fine-tuning the synthetic-pretrained models for just five additional epochs on real VGGSound images yields significant improvements. For fairness, we fine-tuned the real-data baseline for the same number of epochs. These results suggest that the audio and text encoders already capture strong representations, and the main challenge lies in adapting the visual branch to real images. To further prove this, we tested with the AudioCLIP setting by freezing the audio and text branches and fine-tuning only the visual encoder. This achieves comparable gains as the full-finetuning while being over five times faster.

Table 7: **Audio-visual retrieval results.** Category-based audio-visual retrieval accuracy using AudioCLIP (Guzhov et al., 2021) and CAV-MAE Gong et al. (2023) as the backbone. We evaluate both zero-shot retrieval and fine-tuned retrieval. Here, $\mathcal{A}$, $\mathcal{V}$, and $\mathcal{T}$ denote the audio, visual, and text branches, respectively, while ✓ and ✗ indicate whether the encoder for the corresponding modality is frozen or unfrozen during fine-tuning.

| Method | Image type | Branches | | | VGGSound-clean | | VGGSound-test | |
|---|---|---|---|---|---|---|---|---|
| | | $\mathcal{A}$ | $\mathcal{V}$ | $\mathcal{T}$ | $\mathcal{A} \to \mathcal{V}$ | $\mathcal{V} \to \mathcal{A}$ | $\mathcal{A} \to \mathcal{V}$ | $\mathcal{V} \to \mathcal{A}$ |
| AudioCLIP | Real$_{BF}$ | ✗ | ✗ | ✗ | 25.1 | 29.6 | 10.5 | 13.7 |
| | Syn. | ✗ | ✗ | ✗ | 13.1 | 19.7 | 1.9 | 1.1 |
| | Real$_{BF}$ | ✓ | ✓ | ✓ | **26.8** | 32.6 | 11.2 | 14.6 |
| | Syn. | ✓ | ✓ | ✓ | 26.1 | **34.9** | **11.8** | **18.1** |
| | Syn. | ✗ | ✓ | ✗ | 26.3 | 33.1 | 10.7 | 17.3 |
| CAV-MAE | Real$_{BF}$ | ✗ | ✗ | - | 49.4 | 52.4 | 23.5 | 22.5 |
| | Syn. | ✗ | ✗ | - | 36.5 | 22.7 | 17.1 | 7.4 |
| | Real$_{BF}$ | ✓ | ✓ | - | 51.2 | 52.8 | 25.2 | 21.3 |
| | Syn. | ✓ | ✓ | - | **52.8** | **54.1** | **26.4** | **21.8** |

Table 8: **Visual sound localization result.** Performance evaluation of sound localization using EZ-VSL (Mo & Morgado, 2022), trained with either real or synthetized audio. Results are reported as CIoU (Center Intersection over Union) and AuC (Area under the Curve) on the Flickr (Senocak et al., 2018b) and VGG-SS (Chen et al., 2021) datasets.

| method | Audio | Visual | Flickr (Senocak et al., 2018b) | | VGG-SS (Chen et al., 2021) | |
|---|---|---|---|---|---|---|
| | | | CIoU | AuC | CIoU | AuC |
| EZ-VSL (Mo & Morgado, 2022) | Real | Real | 78.3 | 61.0 | 36.3 | 38.4 |
| | Real | Syn. | **81.9** | **62.1** | **36.5** | **38.6** |
| EZ-VSL$_{obj}$ (Mo & Morgado, 2022) | Real | Real | 81.5 | 63.0 | 40.8 | 40.4 |
| | Real | Syn. | **82.7** | **63.5** | **41.1** | **40.5** |

**Visual sound localization.** We further evaluated visual sound localization using EZ-VSL (Mo & Morgado, 2022) as the backbone on the Flickr-SoundNet (Senocak et al., 2018b; 2020) and VGG-SS (Chen et al., 2021) datasets. We choose EZ-VSL as the backbone model because of its simplicity: without heavy augmentation or complex architectural designs, it allows us to more directly and explicitly assess the impact of our synthetic data on model performance. In this task, the model receives an audio clip and an image and is required to identify the region in the image responsible for the sound. Because the self-supervised method relies heavily on alignment—pulling together features from corresponding audio-visual pairs and pushing away unrelated samples in the batch—this setup strongly rewards improved alignment. As shown in table 8, training the model with our augmented dataset yields notable benefits. The cIoU and AuC scores consistently improve on both test sets, especially on Flickr-SoundNet.

## 4.4 ABLATION STUDY

**Synthesizing strategies.** We analyze different image generation strategies using audio-to-image generation as a case study. These include direct generation using audio-conditioned models (Tang et al., 2023) and three variants of our generation pipeline: direct generation with captions, SynCLR's caption engineering (Tian et al., 2023a), and our caption engineering strategy, which places the objects into a reasonable environment. Some qualitative results are shown in appendix A.3.

We assess image generation quality using two performance metrics: audio classification and sound localization. For audio classification, we train a CAVP model and perform linear probing on ESC-50 and FSD-50k. For sound localization, we train an EZ-VSL model and evaluate it on Flickr-SoundNet. As shown in table 9, methods that emphasize accurate alignment between generated images and audio perform well on contrastive learning tasks, likely because they are simpler for the model to learn. In contrast, the SynCLR approach nearly collapses, supporting our hypothesis that overly modified captions can misalign text and audio, resulting in uncorrelated images that hinder audio-visual learning. Nevertheless, the diversity introduced by SynCLR significantly benefits sound localization, possibly because generating objects at various scales captures more real-world

Table 9: **Ablation on different generation strategies.** Performance comparison between different image generation strategies. Results are reported for audio classification accuracy (pretrained with CAVP (Luo et al., 2023)) and sound localization performance (pretrained with EZ-VSL (Mo & Morgado, 2022)). In all cases, the audio data are real.

| Method | Audio CLS. | | Sound Loc. | |
|---|---|---|---|---|
| | ESC-50 | FSD-50k | CIoU | AuC |
| Any2Any (Tang et al., 2023) | 74.0 | 44.0 | 18.1 | 33.3 |
| Direct | 82.0 | 45.1 | 73.4 | 57.8 |
| SynCLR (Tian et al., 2023a) | 69.0 | 40.1 | **83.1** | 61.5 |
| Ours | **82.0** | **45.7** | 81.9 | **62.1** |

variations. Compared with all baselines, our method strikes a balance between correctness and diversity, leading to consistent improvements across tasks.

**Multi-frame.** We compare CAV-MAE training with different input frames in appendix A.4. We find that both multiple synthetic frames setting works the best, as compared to single real/synthetic frames and multiple real frames. For the sampling process for real frames, we uniformly sample 5 frames from the video, and for multiple synthetic frames, we use 5 generated images from the same prompt.

**Balance Real-synthetic ratio.** According to appendix A.5, we perform a grid search on the optimal real and synthetic image/audio ratio, where we find 5% real images and 5% synthetic audio overall give best downstream performance in audio related tasks.

## 5 CONCLUSION

We propose a data generation pipeline that produces well-aligned audio and images using text as a bridge, thereby improving the training of self-supervised audio-visual correspondence models. Additionally, we introduce a filtering method to integrate real and synthetic data, further enhancing data quality. Models trained on our synthetic dataset outperform those trained exclusively on real data across a wide range of downstream tasks. These findings highlight the potential of synthetic data for both scaling up self-supervised models and making training more efficient.

**Limitation and Broader Impact.** All of our methods use popular transformer-based backbones. Moreover, our downstream tasks focus primarily on "high level" tasks that require a semantic understanding of a scene, such as overall scene understanding, making them well-suited to language-based generation. Expanding our experiments to additional datasets could provide further insights. Finally, our model's performance may be influenced by biases in the generative models used during training.

ETHICS STATEMENT

This work uses only publicly available datasets (e.g., VGGSound, AudioSet) and synthetic data generated via large pretrained models, ensuring no private or personally identifiable information is used. Potential risks stem from biases in generative models, which we mitigate through filtering, quality control, and human perceptual evaluation. The human study was conducted under IRB-approved protocols; details will be provided in the camera-ready version.

REPRODUCIBILITY STATEMENT

We provide detailed descriptions of datasets, preprocessing, model architectures, training hyper-parameters, and evaluation protocols in the main text and Appendix. Our experiments were run on 4 NVIDIA A40 GPUs, and we plan to release code, scripts, and processed dataset splits upon acceptance to ensure full reproducibility.

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

# A    IMPLEMENTATION DETAILS

## A.1    HYPERPARAMETER

For contrastive learning, we reproduce the CAVP (Luo et al., 2023), CLAP (Wu* et al., 2023), and AudioCLIP (Guzhov et al., 2021), with a ViT-B/16 (Dosovitskiy et al., 2021) model for visual, a transformer (Vaswani et al., 2017) for text and a HTSAT-tiny (Chen et al., 2022) for audio. For a fair comparison, we developed code for all three methods based on CLAP's code base (Wu* et al., 2023) and used the same set of hyperparameters. For masked autoencoders, we follow CAV-MAE's implementation (Gong et al., 2023) where two ViT-B/16 encoders are trained for both modalities separately except the final block whose parameter is shared between the two modalities. The detailed parameter is shown in table 10 and table 11. For the parameters not mentioned, we use the default parameters contained in the original code base.

Table 10: **Hyper-parameter used to train CLAP, CAVP and AudioCLIP.** For finetune learning rate, three learning rates are searched to find the model with best performance.

| config | Pretrain | Finetune |
|---|---|---|
| optimizer | Adam | Adam |
| base lr | 1e-4 | {1e-4, 2.5e-4, 5e-4} |
| weight decay | 0 | 0 |
| optimizer | $\beta_1, \beta_2 = 0.9, 0.98$ | $\beta_1, \beta_2 = 0.9, 0.98$ |
| batch size | 64 | 32 |
| lr schedule | Cosine | constant |
| epochs | 30 | 10 |

Table 11: **Hyper-parameter used to train CAV-MAE (Gong et al., 2023).**

| config | Pretrain | Finetune |
|---|---|---|
| optimizer | Adam | Adam |
| base lr | 1e-4 | 1e-4 |
| weight decay | 5e-7 | 5e-7 |
| optimizer | $\beta_1, \beta_2 = 0.95, 0.999$ | $\beta_1, \beta_2 = 0.95, 0.999$ |
| batch size | 128 | 40 |
| lr schedule | ReduceLROnPlateau | MultiStepLR |
| warmup epochs | 10 | 0 |
| full epochs | 25 | 10 |
| masking ratio | 75% | N/A |
| contrast loss weight | 0.01 | N/A |
| mae loss weight | 1.0 | N/A |
| label smootht | 0.0 | 0.1 |

## A.2    DETAILED PROMPT

We provide the prompts used to obtain sound descriptions and to generate prompts for image synthesis. First, we obtain SOUND_PLACEHOLDER by prompting the audio captioner with: "Please describe all possible sound events in this audio clip."

Next, we pass SOUND_PLACEHOLDER to a large language model and instruct it to add environmental and background context to the sounding objects using the following prompt: "The following is a sound description: SOUND_PLACEHOLDER. Imagine a plausible environment where this sound could occur. Then, in a single sentence, generate an image description depicting the sound occurring in that environment. Do not introduce any additional sound sources."

## A.3    VISUALIZATION OF DIFFERENT SYNTHESIZE STRATEGIES

We visualize some samples of different synthesize strategies in fig. 3. The images produced by the audio-to-image model and the direct-captions approach exhibit limited diversity; for instance, the generated drum images look nearly identical, sharing the same perspective, background, and

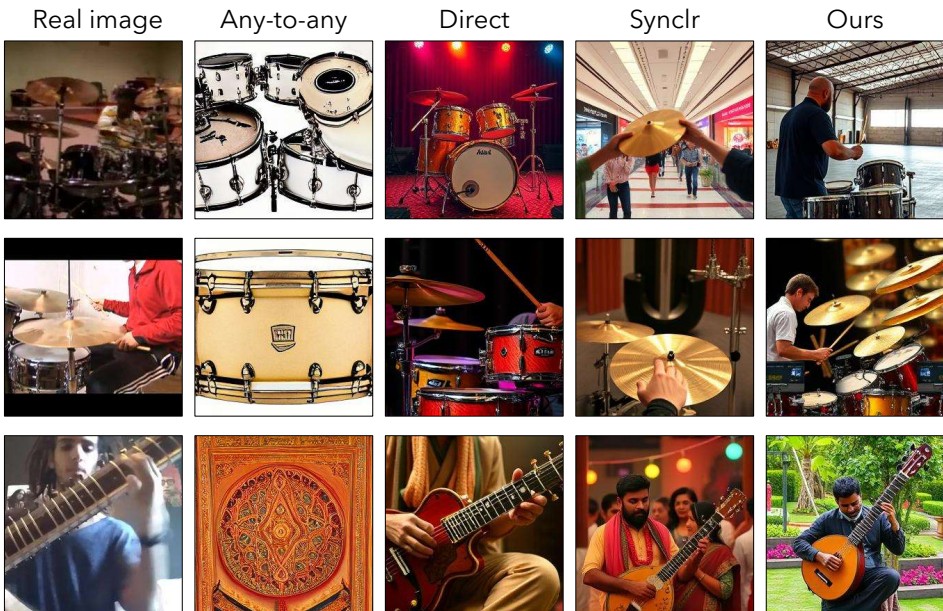

Figure 3: **Qualitative comparison between different generation strategies.** We sample images generated by different methods conditioned on the same audio.

Table 12: **Train with multiple samples.** Performance comparison on audio-visual classification between different frame sampling strategies. CAV-MAE (Gong et al., 2023) is used as backbone.

| Visual | Audio | VGGSound |
|---|---|---|
| Single-real frame | Real | 50.9 |
| Multi-real frames | Real | 51.1 |
| Single-syn frame | Real | 51.5 |
| Multi-syn frames | Real | **51.9** |

lighting. Additionally, audio-to-image generation can lead to missed objects (e.g., failing to generate a guitar), since audio does not always convey sufficient visual information. Although prompt-engineered generation mitigates the low-diversity issue, SynCLR's method—which makes significant changes to captions and introduces extra elements—sometimes produces unrealistic or inconsistent images.

### A.4 DETAILED ABLATION OF FRAME NUMBERS

We extend our experiments to train with multiple frames, using CAV-MAE (Gong et al., 2023) as the backbone for evaluation. For the real data, we uniformly sample five frames per video. For the synthetic dataset, we generate five images from the same prompt. During training, the model randomly selects one frame as the image input. As shown in table 12, introducing additional frames significantly boosts the real-data model's performance by reducing the chance of choosing a suboptimal frame. Nevertheless, our approach still benefits from multiple frames, outperforming the version trained entirely on real data.

### A.5 DETAILED ABLATION OF BALANCING REAL AND SYNTHETIC DATA

We investigate the optimal ratio of real and synthetic data, as shown in fig. 4. We use CAVP as the backbone and evaluate performance via linear probing on ESC-50 and FSD-50k. First, we explore adding real images to a dataset comprising real audio paired with synthetic images. Specifically, we replace synthetic images with the highest-scoring real images according to their audio-visual alignment scores. A small proportion of real images yields slight performance gains by improving overall dataset quality and better matching the real-data distribution, thus mitigating out-of-distribution issues. However, exceeding a certain threshold of real images degrades performance: the growing

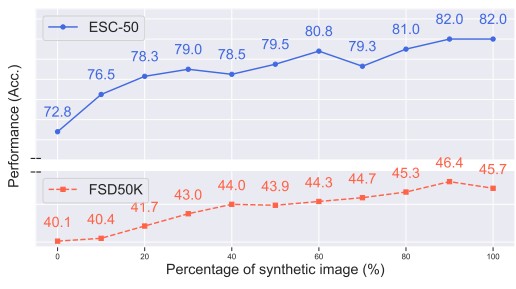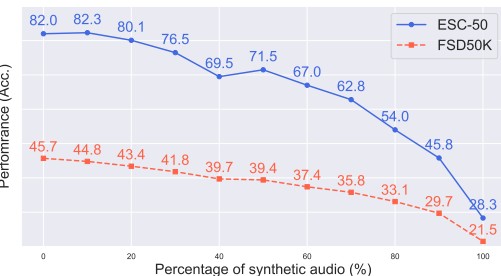

(a) Different synthetic image ratio with real audio

(b) Different synthetic audio ratio with synthetic image

Figure 4: **Data mixture ratio.** We show how varying the ratio of synthetic image and audio affects the downstream tasks' performance on two audio classification benchmarks, ESC50 and FSD50k. Performance generally improves as more synthetic images are added and suffers when more synthetic audio is added.

disparity between real and synthetic distributions complicates training and reintroduces many misaligned examples.

Next, we analyze increasing the amount of synthetic audio by replacing samples with low alignment scores. Adding a small volume of synthetic audio alleviates data collapse. However, excessive synthetic audio skews the training distribution away from the test distribution, leading to a marked drop in performance.

Based on these observations, we conduct a fine-grained search to identify the best ratio of real to synthetic data for both modalities. As shown in table 13, using 5% real images and 5% synthetic audio yield the best overall results.

Table 13: **Combination ratios.** We explore different ratios for audio representation learning.

(a) Linear probing result on ESC50

|  | 90% Syn Image | 95% Syn Image | 100% Syn Image |
|---|---|---|---|
| 0% Syn Audio | 82.0 | 83.3 | 82.0 |
| 5% Syn Audio | 82.8 | **83.8** | 82.3 |
| 10% Syn Audio | 83.3 | 81.8 | 82.3 |

(b) Linear probing result on FSD50k

|  | 90% Syn Image | 95% Syn Image | 100% Syn Image |
|---|---|---|---|
| 0% Syn Audio | **46.4** | 45.8 | 45.7 |
| 5% Syn Audio | 45.4 | 46.2 | 45.0 |
| 10% Syn Audio | 44.6 | 44.9 | 44.8 |

## A.6 CROSS-MODAL MATCHING MODELS

We evaluated the performance of various cross-modal matching models for filtering, where these models compute similarity scores between audio and image modalities (table 14). The datasets filtered using these similarity scores were subsequently used to pretrain the CAVP model, and the pretrained model's performance was assessed on audio classification tasks. For methods like LLAVA (Liu et al., 2023) and CLIP score (Hessel et al., 2022) which only acquire visual and text as input, we directly use the label for filtering as this maximizes the correctness. For ImageBind (Girdhar et al., 2023), we use audio and visual as input. We found that large-language-model-based models like LLAVA encounter hallucination issues, which leads to a low score. CLIP also performs worse than ImageBind, as ImageBind can directly calculate the similarity between audio and visual. Thus, we chose ImageBind for filtering's backbone model.

Table 14: **Performance of different Cross-Modal Matching Models for filtering** Experiments are conducted on FSD-50k audio classification task with pretrained model using CAVP as the backbone.

| Filter Method | modalities | FSD-50k |
|---|---|---|
| Pure synthetic | N/A | 45.7 |
| CLIP score | Visual-text | 44.7 |
| LLAVA | Visual-text | 40.9 |
| ImageBind | Audio-visual | **46.2** |

### A.7 AUDIO CAPTIONING MODEL

We evaluated the performance of several audio captioning models, including Salmonn (Tang et al., 2024), Qwen (Bai et al., 2023), and Qwen-v2 (Yang et al., 2024), using the downstream task performance of CLAP as the evaluation metric (table 15). Based on these evaluations, we selected Salmonn as the final model due to its optimal balance of computational efficiency and classification performance.

Table 15: **Comparing audio captioner with CLAP.** Salmonn gives the best performance on FSD-50k audio classification.

| Text | FSD-50k |
|---|---|
| Real label | 46.1 |
| Salmonn (Tang et al., 2024) | **46.3** |
| Qwen (Bai et al., 2023) | 45.1 |
| Qwenv2 (Yang et al., 2024) | 46.2 |

### A.8 SYNTHETIC IMAGE GENERATION AND EDITING

We explored the most effective methods for generating synthetic images by evaluating their impact on downstream audio classification tasks, with results summarized in table 16. The performance of each method was measured using CAVP as the evaluation metric. In addition to generating images from scratch, we tested image editing methods, such as SDEdit (Meng et al., 2022), which attempts to add the sounding object directly to existing images. However, our experiments revealed that these editing methods consistently underperformed compared to images generated from scratch.

Among the tested methods, the Flux-schnell model has the best performance, producing the highest accuracy on audio classification tasks while maintaining the fastest image generation speeds. This balance of speed and quality makes the distilled Flux model the most suitable choice for our synthetic image generation pipeline.

Table 16: **Comparison of Image Generation and Editing Methods Using CAVP.** For SDEdit, Stable Diffusion 1.5 was used as the backbone. Generation speed was measured using a single NVIDIA A40 GPU with default hyperparameters. The distilled Flux model achieves the best performance on FSD-50k audio classification while offering the fastest generation speed.

| Method | Speed (s/image) | FSD-50k |
|---|---|---|
| Real | - | 40.1 |
| SDEdit (Meng et al., 2022) | - | 40.3 |
| Stable-1.5 (Rombach et al., 2022) | 4 | 43.5 |
| Stable-2.1 (Rombach et al., 2022) | 5 | 43.8 |
| Stable-XL (Podell et al., 2023) | 5 | 45.2 |
| Flux-schnell (Lipman et al., 2023) | 1 | **45.7** |

## B   EXPERIMENT RESULTS FOR VGGSOUND AUDIO CLASSIFICATION

In table 17, we also report the pretrained CAVP model's performance on VGGSound audio classification. Similar to the results in table 3, the model trained on our synthetic dataset outperforms the one trained on real data.

## C   FILTERING SAMPLES

We demonstrate the filtered results with the top three highest scores and the lowest scores in fig. 5. For the low-similarity samples, the first two incorrectly match the label, and the third one has significant background noise.

Table 17: **VGGSound audio classification.** CAVP pretrained on real vs. synthetic VGG-SS, evaluated on VGGSound's audio.

| Image data | VGGSound audio cls. |
|---|---|
| Real$_{BF}$ | 31.4 |
| Real$_{MF}$ | 31.7 |
| Syn | **35.0** |

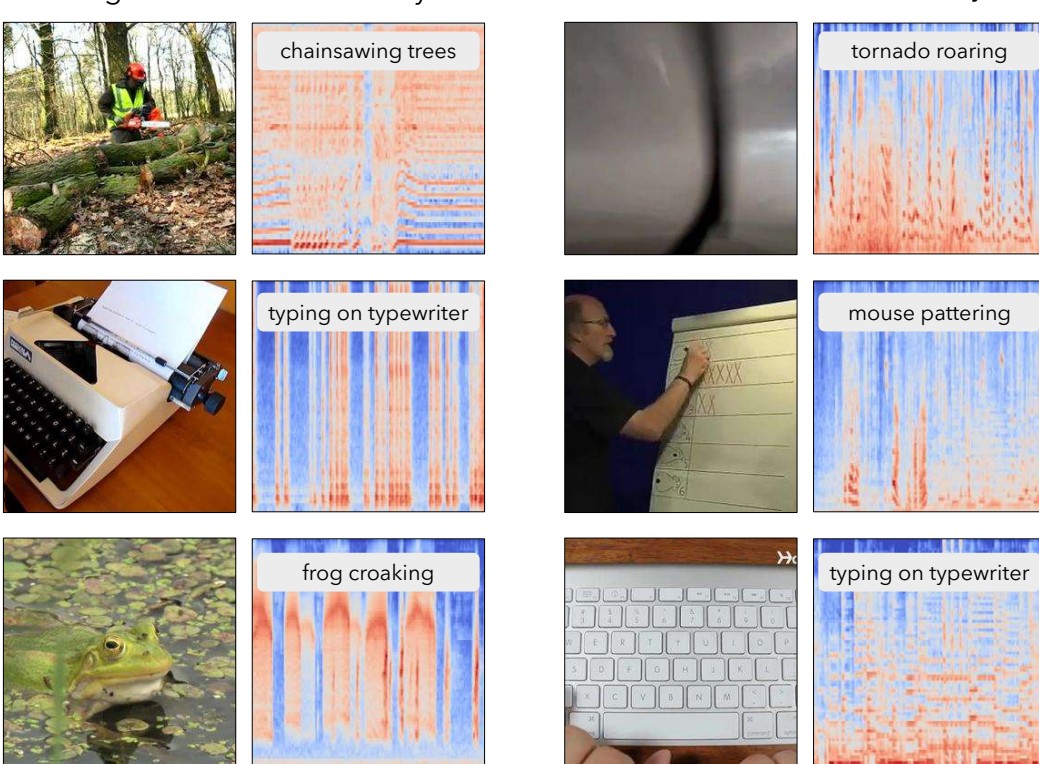

Figure 5: **Samples with high and low audio-visual similarity score.** Left are audio-visual samples with the highest ImageBind similarity, and right are samples with the lowest similarity.

## D MORE EXAMPLES

We provide more examples of our generated audio-visual pairs in fig. 6 and fig. 7. Example audio clips are also included in the video.

## E HUMAN STUDY

To elaborate on our human study, we present the website UI in fig. 8 and provide a video demonstrating how it works. Essentially, users choose between synthetic samples and real samples to indicate which one is better or if both are good or bad. Samples given to each user are different and randomly chosen from the dataset. The order of real and synthetic samples is also randomly shuffled. We get totally about 100 responses for both the audio and image data.

Further, we conduct some experiment to show there is a correlation between the human study and the automatic evaluation pipeline. However, the two evaluation settings differ significantly: automatic evaluation measures semantic distance and classifies results as favoring either real or synthetic data, while the human study includes two additional options: both good and both bad. To ensure a fair comparison, we calculated the Spearman coefficient after excluding samples from these two additional categories. The final value is 0.21 ($p = 0.035$), indicating a meaningful positive correlation.

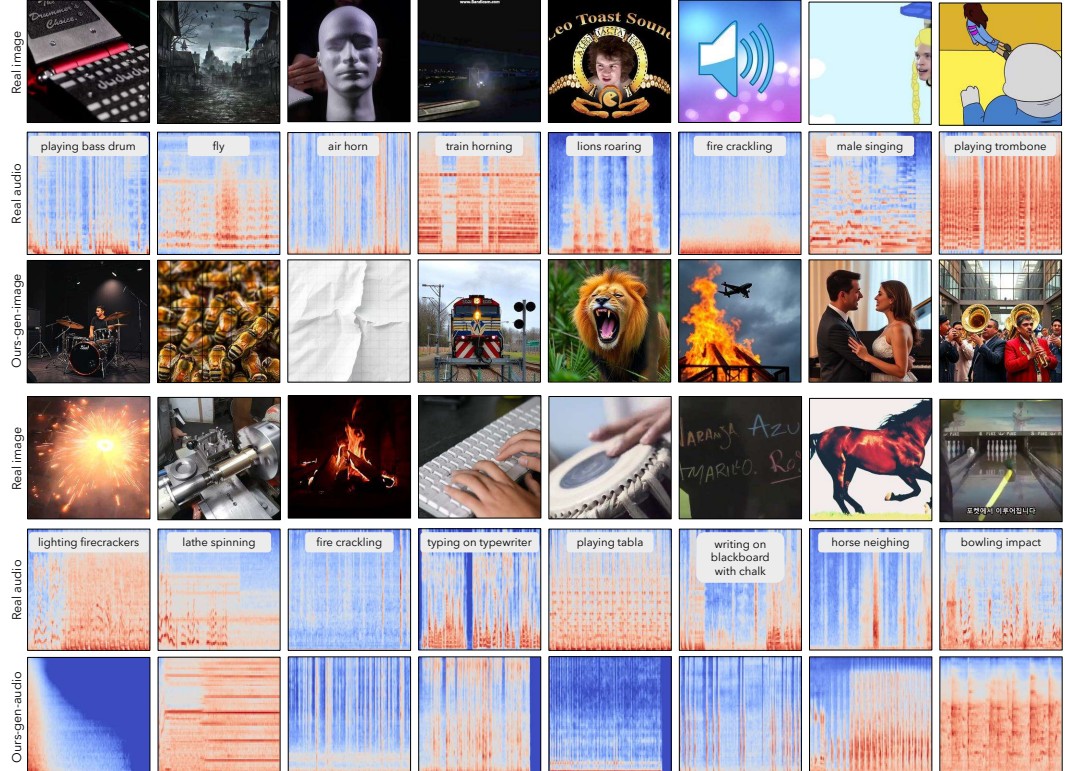

Figure 7: **More generated audio examples.**

## F  LLM STATEMENT

We use ChatGPT for revising the grammar of the writing.

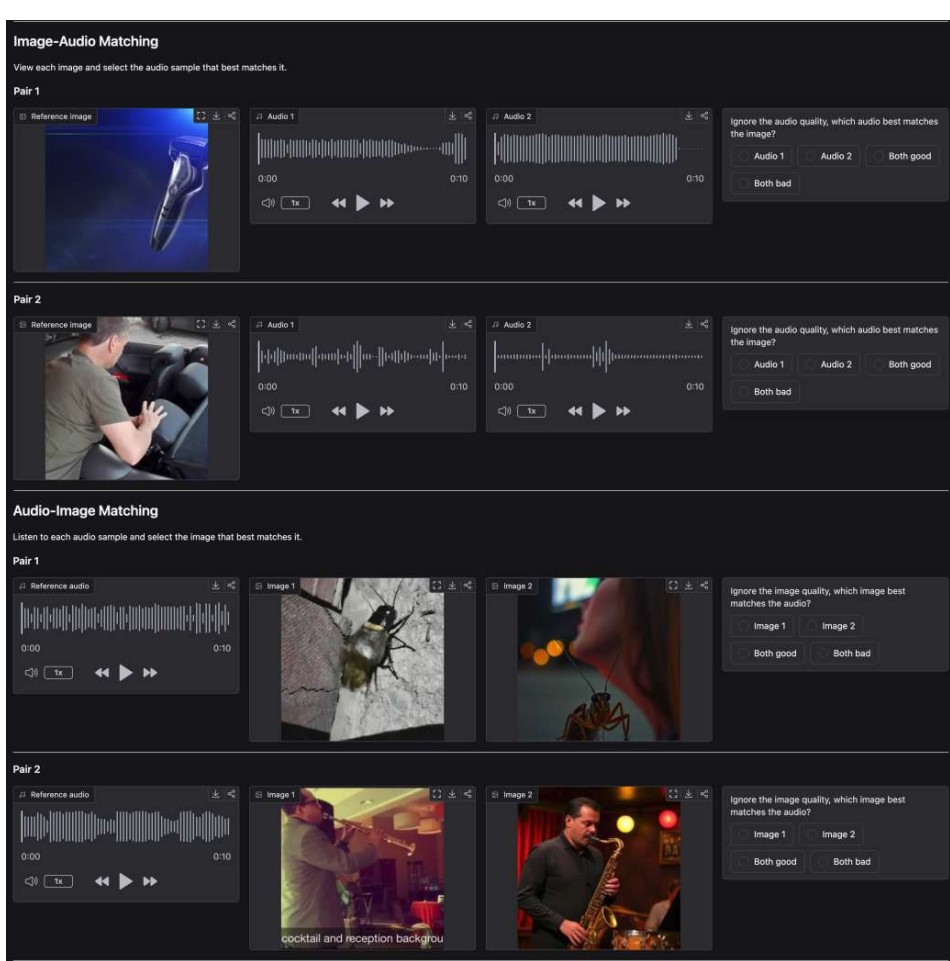

Figure 8: **User study UI.**

