# OpenReview forum: "Bridging Audio-Visual Semantics with Language-Guided Synthesis"
_ICLR.cc/2026/Conference — Submitted to ICLR 2026_

### Official Review · Reviewer_sxkD · 2025-10-29

**Soundness:** 2
**Presentation:** 2
**Contribution:** 2
**Rating:** 4
**Confidence:** 4

**Summary:**

The paper proposes a data generation framework that uses language as an intermediate bridge to create semantically consistent audio-visual pairs, solving the problem of misaligned or low-quality data that often degrades multimodal learning models. By generating synthetic images conditioned on audio and synthetic audio conditioned on images, the method repairs corrupted real-world data and generates a highly aligned dataset. The primary contribution is demonstrating that features learned using representations trained on this resulting mixed dataset which combines filtered real data with the high-quality synthetic data outperform models trained solely on the original real data across tasks like audio classification, cross-modal retrieval, and visual sound localization. Human and automatic evaluations confirm that the synthesized audio-visual pairs are more semantically aligned than those found in the original dataset.

**Strengths:**

- The framework made progress in addressing the issue of misaligned or mismatched audio-visual signals in real-world datasets by using cross-modal generative models to replace corrupted data. Features learned using the resulting mixed dataset (real and synthetic data) lead to better results across a variety of downstream tasks, including audio classification, audio-visual retrieval, and visual sound localization
- The method introduces a filtering process and mixing strategy that combines the best-aligned real data with synthetic data, which is found to outperform models trained purely on either real or synthetic data
- The synthetic data is suitable for a wide range of representation learning algorithms, including contrastive learning methods (like CAVP) and reconstruction-based methods (like CAV-MAE)

**Weaknesses:**

- The method's core technical strategy involves chaining together several existing, powerful models. While the application of this pipeline to repair audio-visual mismatch is novel, the individual components and their mechanisms are off-the-shelf. Moreover, Language-as-a-Bridge is an established concept. models like ImageBind align multiple modalities by connecting them to frozen vision–language embeddings, and LG-CAV-MAE employs captioning to associate signals with text. I think authors should better clarify that the novelty lies in leveraging this bridge to generate new, corrected data rather than simply aligning existing data.
- The framework uses individual video frames as the visual signal, rather than generating or processing full video sequences, a choice made because generating full video is a challenging open problem. This limits the model's ability to utilize the temporal visual information available in a full video clip.
- The reported experiments focus mainly on high-level downstream tasks, such as overall scene understanding (e.g., classification and retrieval), which are well-suited to the language-based generation approach employed.

**Questions:**

- Is a fine-grained search of the balance data mixing ratio necessary for every new dataset used for pretraining?
- The caption rewriting strategy was carefully developed to avoid hallucination by the language model and semantic misalignment by altering only the environment and perspective. How was the threshold for modifying captions determined, and what specific metrics or internal evaluations were used to quantify the risk of semantic collapse before choosing the final, constrained approach?
- What specific limitations or performance degradations would the authors expect if this language-guided framework were applied to more fine-grained, low-level audio analysis tasks (e.g., detecting quick transients or specific frequency changes) where language descriptions might lack the necessary acoustic detail?

---

> ### Author Response · Authors · 2025-11-22
> **Response to Reviewer sxkD (Part1)**
>
> We sincerely appreciate your careful and thorough review. Below we provide point-by-point responses to your comments.
>
> > W1: The method's core technical strategy involves chaining together several existing, powerful models. While the application of this pipeline to repair audio-visual mismatch is novel, the individual components and their mechanisms are off-the-shelf. Moreover, Language-as-a-Bridge is an established concept. models like ImageBind align multiple modalities by connecting them to frozen vision–language embeddings, and LG-CAV-MAE employs captioning to associate signals with text. I think authors should better clarify that the novelty lies in leveraging this bridge to generate new, corrected data rather than simply aligning existing data.
>
> Thanks for the question.  We agree that ImageBind and LG-CAV-MAE use text to tie modalities together. Our novelty lies in how this bridge is used:
>
> - Prior work typically aligns existing signals in representation space.
>
> - We instead use language as an intermediate specification to generate new signals, i.e., we use captions to synthesize the missing modality and thereby repair misaligned pairs at the data level.
>
> This makes our contribution primarily a dataset contribution and how to create it: we show that repairing noisy AV data via language-guided synthesis consistently improves multiple downstream models and tasks, without changing their architecture. We will emphasize this angle in the introduction and related work.
>
> > W2: The framework uses individual video frames as the visual signal, rather than generating or processing full video sequences, a choice made because generating full video is a challenging open problem. This limits the model's ability to utilize the temporal visual information available in a full video clip.
>
> We intentionally adopt single frames for two reasons:
>
> -  Most AV representation baselines we build upon (e.g., AudioCLIP) are designed for single-frame inputs, which makes comparison cleaner.
>
> - High-quality video generation from audio/text is still an open and computationally heavy problem; incorporating it would conflate our main message with video-generation quality.
>
> We acknowledge that temporal information is important; to partially address this, we include a multi-frame ablation (**Table 9**) showing that even when multiple frames are used, training on our synthetic data still yields gains over training on real data.
>
> > W3: The reported experiments focus mainly on high-level downstream tasks, such as overall scene understanding (e.g., classification and retrieval), which are well-suited to the language-based generation approach employed.
>
> Our target is at learning better high level representations, so our main tasks like audio classification, AV retrieval, follow standard practice in AV representation learning. However, we still find some boost in the  fine-grained, low-level tasks like sound localization when using our synthetic data than the real ones.

---

> ### Author Response · Authors · 2025-11-22
> **Response to Reviewer sxkD (Part2)**
>
> > Q1: Is a fine-grained search of the balance data mixing ratio necessary for every new dataset used for pretraining?
>
> In practice, we find that a moderate range of mixing ratios (e.g., synthetic:real between 0.5:1 and 1:1) already yields improvements; exhaustive per-dataset tuning is not strictly necessary. In fact, even using only synthetic data outperforms using only real data in many settings, and mixing mostly provides an extra boost by better matching distributional details of the downstream test sets.
>
> > Q2: The caption rewriting strategy was carefully developed to avoid hallucination by the language model and semantic misalignment by altering only the environment and perspective. How was the threshold for modifying captions determined, and what specific metrics or internal evaluations were used to quantify the risk of semantic collapse before choosing the final, constrained approach?
>
> Our caption-rewriting process is intentionally conservative: the LLM is instructed to modify only environmental details, stylistic elements, and other non-core attributes while preserving the main event described in the audio. In preliminary experiments, we first designed and validated our prompt templates using toy examples from AudioCaps before scaling up. During this phase, we found that both overly large and overly small modifications to the original captions led to degraded performance on downstream tasks. These findings motivated the constrained rewriting strategy adopted in the paper.
>
> > Q3: What specific limitations or performance degradations would the authors expect if this language-guided framework were applied to more fine-grained, low-level audio analysis tasks (e.g., detecting quick transients or specific frequency changes) where language descriptions might lack the necessary acoustic detail?
>
> We agree that our language-guided synthesis may not preserve those fine-grained audio details. In such cases, there is a trade-off between improving semantic alignment and potentially losing fine-grained acoustic information. However, we also argue that our work aims to build high quality data through multi-modal cues, and these details cannot be found in either language or image.

---

### Official Review · Reviewer_vJDQ · 2025-10-29

**Soundness:** 3
**Presentation:** 3
**Contribution:** 2
**Rating:** 4
**Confidence:** 5

**Summary:**

The paper aims to address audio-visual misalignment in web video by replacing the mismatched modality with language-guided synthetic counterparts (audio↔image) and mixing them with the best-aligned real pairs via embedding-space filtering. Train standard audio-visual self-supervised learners on this “repaired” corpus. After using these synthetic data together with real data, different methods achieve consistent gains on audio classification, AV retrieval, and visual sound localization.

**Strengths:**

1. The paper is generally well-written and the core idea is simple and seems effective.

2. Consistent gains are seen across multiple tasks (Audio CLS (ESC-50 / FSD-50K / Urban8K), ) after using the synthesized data for models training, and helps all models to improve (CAV-MAE, CAVP, AudioCLIP).

3. The paper also compares different synthetic strategies and shows their advantage over the others in the ablations on audio cls and sound localization task.

**Weaknesses:**

1. The synthetic data is generated using rather weak models, why not use stronger frontier-models like these omni models like Qwen-Omni, MiniCPM-o-2.6, or even like Gemini / GPT ones. The paper should ablate different model's impact on synthetic data quality.

2. The benchmark evaluation lacks an important dataset, which is AudioSet, they can be used for both audio-visual classification and retrieval, the paper should definitely include the results. VGGSound dataset is still considered very small for these tasks.

3. Lack of important citations related to audio-visual representation learning, e.g., [1] and [2].

4. The proposed method is still constrained within the scope of semantic alignment, if extend to video temporal alignment, then the proposed pipeline would be considered as limited.


[1] From vision to audio and beyond: A unified model for audio-visual representation and generation, ICML 2024.

[2] CAV-MAE Sync: Improving Contrastive Audio-Visual Mask Autoencoders via Fine-Grained Alignment, CVPR 2025.

**Questions:**

See weaknesses above.

---

> ### Author Response · Authors · 2025-11-22
> **Response to Reviewer vJDQ**
>
> We sincerely appreciate your careful and thorough review. Below we provide point-by-point responses to your comments.
>
> >W1: The synthetic data is generated using rather weak models, why not use stronger frontier-models like these omni models like Qwen-Omni, MiniCPM-o-2.6, or even like Gemini / GPT ones. The paper should ablate different model's impact on synthetic data quality.
>
>  We agree that the choice of generator is important. Our current experiments use open models that balance generation quality with feasible compute. We also ran an ablation across several image generators in the supplement **Table 13** and **Table 14** and observed that higher-quality generators generally produce better-performing synthetic data, but the relative improvements from adding synthetic data remain consistent across generators.
>
> Using frontier omni-models (Qwen-Omni, MiniCPM-o, Gemini, GPT-4o) at scale is currently limited by API and compute constraints; however, our pipeline is model-agnostic and would likely benefit from them.
>
> > W2: The benchmark evaluation lacks an important dataset, which is AudioSet, they can be used for both audio-visual classification and retrieval, the paper should definitely include the results. VGGSound dataset is still considered very small for these tasks.
>
> We just finished generating a synthetic version of AudioSet and train a CAV-MAE model on it. Due to dead links in the original dataset, our synthetic set contains 1.7 M image–audio pairs. Even though, our model still show a better performance than the model trained with full. By finetuning with the same dataset and hyper-parameter, on audio classification of  VGGSound,  our model achieves an accuracy **58.9**, compared to the **58.2** that trained with the real one.  For the audio-visual classification, we get **62.5** while the real baseline achieve **61.9**.
>
> >W3: Lack of important citations related to audio-visual representation learning
>
> Thank you for pointing out the missing references. We will add and discuss:
> “From vision to audio and beyond: A unified model for audio-visual representation and generation” (ICML 2024), and   “CAV-MAE Sync: Improving Contrastive Audio-Visual Mask Autoencoders via Fine-Grained Alignment” (CVPR 2025),
> and position our contribution as complementary: our synthetic dataset can, in principle, be used as training data for these more advanced architectures as well.
>
>
> >W4: The proposed method is still constrained within the scope of semantic alignment, if extend to video temporal alignment, then the proposed pipeline would be considered as limited.
>
> We intentionally adopt single frames for two reasons:
>
> -  Most AV representation baselines we build upon (e.g., AudioCLIP) are designed for single-frame inputs, which makes comparison cleaner.
>
> - High-quality video generation from audio/text is still an open and computationally heavy problem; incorporating it would conflate our main message with video-generation quality.
>
> We acknowledge that temporal information is important; to partially address this, we include a multi-frame ablation (**Table 9**) showing that even when multiple frames are used, training on our synthetic data still yields gains over training on real data.

---

### Official Review · Reviewer_ExQR · 2025-10-30

**Soundness:** 2
**Presentation:** 3
**Contribution:** 2
**Rating:** 4
**Confidence:** 4

**Summary:**

This work proposes a method to resolve the issue of mismatched audio–visual signals using a cross-modal generative model. Specifically, either an audio caption or an image caption is first predicted from the original signal using off-the-shelf models, refined through a large language model, and then fed into a text-to-audio or text-to-image generative model to synthesize a semantically matched pair.
To evaluate the effectiveness of this synthesis approach, the authors learn several audio–visual representations using the synthesized (or combined) datasets and apply these representations to various downstream tasks, such as audio classification and visual sound localization. Experimental results show that incorporating synthetic data during training consistently improves performance on downstream tasks compared to using only real datasets.

**Strengths:**

- The issue of modality mismatch is an important challenge, and approaching it through a generative model is a promising direction.

- The empirical experiments are comprehensive, covering diverse downstream tasks to validate the proposed approach.

- The ablations are well designed to understand the effectiveness of the method.

- The writing is clear and easy to follow.

**Weaknesses:**

- While the results demonstrate that synthetic data improves downstream performance, it remains unclear how well the combined real–synthetic dataset generalizes to more recent or stronger baseline models. For example, methods such as “Sound Source Localization is All about Cross-Modal Alignment, ICCV 2023” already surpass the quantitative results shown in Table 6 without using synthetic data. Including additional experiments on more recent models would help verify the generalizability and lasting impact of the proposed approach of generating a synthetic dataset and using it for downstream tasks.

- Some aspects of the method description lack clarity and reproducibility, such as the design of prompts for caption refinement or diversity control. Including example prompts or generation templates would make the method more transparent and reproducible.

- It is also unclear how detailed and diverse the generated images are. How do the authors control for the noisiness, bias, or stylistic artifacts introduced by the text-to-image or text-to-audio generative models?

- Audio classification results on VGGSound are not reported. While the paper includes audio–visual classification on the VGGSound test set, a direct audio-only classification experiment (similar to Table 3) would provide a more complete understanding of the learned audio representations.

- The coverage of long-tail or rare events using synthetic data is not analyzed. Since many mismatched scenarios occur in rare or underrepresented events, synthetic generation could be particularly beneficial here. A detailed analysis of which categories or domains show performance degradation—and where synthetic augmentation provides improvement—would provide valuable insight into the strengths and limitations of the approach.

**Questions:**

### Questions and Suggestions for Further Experiments
To more thoroughly validate the contribution, the following additional experiments and analyses are recommended:

- Evaluating generalization to more recent or stronger models will confirm the consistent effectiveness of synthetic data augmentation.

- Including audio classification results on the VGGSound test set will complement the existing audio–visual experiments.

- Conducting an error analysis on the data synthesis pipeline, including potential failure cases of the generative models and their effects on downstream performance.

- Analyzing category- or domain-level performance, highlighting where the current model struggles and how synthetic data alleviates these weaknesses, will support the effectiveness of the synthetic dataset.

### Minor Questions and Suggestions

- L279, L379, L408: Use \citep{} instead of \cite{} for consistency with citation style.

- Please capitalize “Figure” and “Table” throughout the paper (e.g., L313: fi.g2 → Figure 2).

- In Table 7, bold the ESC-50 result for the Direct method for readability.

- Provide more details on prompt design for both image and audio generation. For instance, how are environmental variations introduced? Are there structured templates, constraints, or diversity prompts used?

-For tasks that are audio‐only (e.g., ESC-50 classification), how does adding synthetic images help? Do you see gains purely because the audio is unchanged, but training data context changes? Please clarify the mechanism.

---

> ### Author Response · Authors · 2025-11-22
> **Response to Reviewer ExQR**
>
> We sincerely appreciate your careful and thorough review. Below we provide point-by-point responses to your comments.
>
> > W1: While the results demonstrate that synthetic data improves downstream performance, it remains unclear how well the combined real–synthetic  dataset generalizes to more recent or stronger baseline models. For example, methods such as “Sound Source Localization is All about Cross-Modal Alignment, ICCV 2023” already surpass the quantitative results shown in Table 6 without using synthetic data. Including  additional experiments on more recent models would help verify the generalizability and lasting impact of the proposed approach of  generating a synthetic dataset and using it for downstream tasks.
>
> We compare our constructed synthetic dataset against real data to evaluate how it benefits the sound source localization task. We choose EZ-VSL as the backbone model due to its simplicity: without heavy augmentation or complex architectural designs, it allows us to more directly and explicitly assess the impact of our synthetic data on model performance.
>
>
> > W2: Some aspects of the method description lack clarity and reproducibility, such as the design of prompts for caption refinement or diversity control. Including example prompts or generation templates would make the method more transparent and reproducible.
>
> Following is the major prompt we are using:
> For the audio captioner, we query $SOUND$ by "Please describe all possible sound events in this audio clip."
>
> For get the prompt for generation, we query the LLM by: The following is a sound description: $SOUND$. Imagine a possible environment where this sound is happening. Then within one sentence, generate an image description where the sound occurs in such an environment. Do not add any other sounding objects.
>
> > W3: It is also unclear how detailed and diverse the generated images are. How do the authors control for the noisiness, bias, or stylistic artifacts introduced by the text-to-image or text-to-audio generative models?
>
> Thanks for the suggestion. We further assess the diversity and quality of the generated samples by computing the Vendi score on a randomly selected subset of 3,000 images. Images generated without our prompt refinement achieve a Vendi score of **15.9**, whereas images generated with our prompt template reach **21.9**, indicating higher diversity. You can also check **Figure 3** for qualitative results.
>
> For control, as intermediate stages of generation are difficult to control directly, we instead adopt an end-to-end evaluation strategy: we empirically select prompt templates that yield the best downstream performance on some small scale dataset like AudioCaps. This ensures that any potential noisiness, bias, or stylistic artifacts introduced by the generative models are mitigated based on their actual impact on task performance rather than heuristic assumptions.
>
>
> > W4: Audio classification results on VGGSound are not reported. While the paper includes audio–visual classification on the VGGSound test set, a direct audio-only classification experiment (similar to Table 3) would provide a more complete understanding of the learned audio representations.
>
> We evaluate VGGSound audio using the same pretraining and testing setup as in Table 3. Under this setting, CAVP (middle-frame training) achieves **31.4**, CAVP (best-frame training) achieves **31.7**, and our method achieves **35.0**. This trend is also quite aligned with the other dataset. We will soon update the other numbers in the revised paper.
>
>
> >W5: The coverage of long-tail or rare events using synthetic data is not analyzed. Since many mismatched scenarios occur in rare or underrepresented events, synthetic generation could be particularly beneficial here. A detailed analysis of which categories or domains show performance degradation—and where synthetic augmentation provides improvement—would provide valuable insight into the strengths and limitations of the approach.
>
> We do find a few categories where the original audio-image pair fails, such as sounding objects don't exist in the image. For example, as shown in **Figure 2**, there is few entire black corresponding image with the audio of a thunder in VGGSound. Our synthetic data can help a lot making these scenes having better pairs. For the limitation, we think as we are heavily relying the pretrained models for give correct captions. So if the sound is faint or noisy, the captioner may not give correct caption and this may degrade our method's performance.

---

### Official Review · Reviewer_YqAF · 2025-10-30

**Soundness:** 2
**Presentation:** 2
**Contribution:** 2
**Rating:** 4
**Confidence:** 4

**Summary:**

The paper proposes a new method for audio-visual representation learning. The core idea is to "repair" or "enhance" misaligned parts in the original audio-visual data through a language-guided cross-modal generation model (e.g., text-to-image, text-to-audio) to improve the effectiveness of self-supervised learning. The authors constructed a hybrid dataset containing both synthetic and real data and validated its effectiveness on several downstream tasks (such as audio classification, audio-visual retrieval, sound source localization).

**Strengths:**

By using text as a bridge to generate images from audio or audio from images, the method repairs misalignments in the original data. The effectiveness of the synthetic data was verified through human perception evaluation, automated semantic similarity evaluation, and comparisons with various representation learning methods. The method's effectiveness was validated on multiple benchmark datasets, with models trained on synthetic data outperforming those trained only on real data.

The paper has a certain degree of originality. While using synthetic data for visual representation learning has been studied, applying it to audio-visual alignment and using language-guided generation to repair misaligned data is a novel and practically valuable direction. The expression and layout of the paper still have room for improvement, such as the complex representation of evaluation mechanisms and the data construction pipeline in the charts. The paper holds a certain level of significance.

**Weaknesses:**

1. Further assessment of data quality is needed. For example, existing contrastive learning models could be used to evaluate semantic cosine similarity, especially in comparison with datasets constructed by other methods, such as Ex-MCR's retrieval-based dataset construction approach.

2. There is a lack of direct performance comparisons with other data-driven methods, such as ImageBind and LanguageBind.

Ex-MCR: https://proceedings.neurips.cc/paper_files/paper/2024/file/a71df365f872a39e58475f1fa7950879-Paper-Conference.pdf

ImageBind : https://arxiv.org/abs/2305.05665

LanguageBind : https://arxiv.org/abs/2310.01852

**Questions:**

1. I noticed the paper's discussion on the distributional inconsistency between generated data and real data. Could you elaborate on the data construction pipeline used during fine-tuning? Specifically, what are the sources of the images, text, and audio in the fine-tuning dataset?

2. The current generation models mentioned in the paper, such as FLUX and Stable Audio Open, also rely on contrastive learning models for prompt understanding (these models include pre-trained contrastive learning models in their structures). Essentially, can the paper's method be considered as a distillation and integration of semantic modality alignment within existing generative models? If so, can the paper's method be compared in terms of model performance with methods like Ex-MCR that also integrate existing contrastive learning models?

FLUX: https://arxiv.org/abs/2210.02747

Stable Audio Open: https://arxiv.org/abs/2407.14358

---

> ### Author Response · Authors · 2025-11-22
> **Response to Reviewer YqAF (Part 1)**
>
> We sincerely appreciate your careful and thorough review. Below we provide point-by-point responses to your comments.
>
> >W1: Further assessment of data quality is needed. For example, existing  contrastive learning models could be used to evaluate semantic cosine  similarity, especially in comparison with datasets constructed by other  methods, such as Ex-MCR's retrieval-based dataset construction approach.
>
> We agree that quantitative assessment of data quality is crucial. Our current **Section 4.2** already performs a two evaluation:
>
> - Human study: We ask annotators to judge which audio–image pair is more semantically aligned. The reported numbers represent the sum of “both good” and “prefer synthetic”, hence directly measuring perceived semantic consistency rather than only image quality.
>
> - Automatic semantic similarity: We compute cosine similarity between audio and image captions using Sentence-BERT embedding, exactly following the spirit of the reviewer’s suggestion to use pretrained models to measure semantic alignment. We find that both audio-to-image and image-to-audio synthesized pairs achieve higher semantic similarity than the original real pairs.
>
> We did not use CLIP or ImageBind themselves as evaluators, because those encoders are trained on large-scale real image–text/audio–text datasets. As a result, they tend to over-score real images that lie closer to their training distribution, making the comparison between real and synthetic data biased.
>
> Regarding Ex-MCR, it constructs retrieval-based pseudo pairs by aligning multiple pretrained embedding spaces, whereas our method directly generates new cross-modal content, providing true synthetic audio–image pairs rather than re-paired existing samples.  Thus, Ex-MCR is a representation-alignment technique, while our method is a data-generation framework aimed at creating semantically grounded training pairs, so we think these two approaches  are complementary rather than comparable. We will include this discussion within the related works part.
>
> > W2: There is a lack of direct performance comparisons with other data-driven methods, such as ImageBind and LanguageBind.
>
> Our focus is on well aligned dataset construction: using language-guided synthesis to repair misaligned audio–visual pairs, so that existing representation learning methods (CAV-MAE, CAVP, AudioCLIP, etc.) can benefit without any architectural changes. In contrast, ImageBind and LanguageBind are high-capacity unified models trained with very different objectives, modalities, and data scales. So we think these are not in the same scope of comparison.

---

> ### Author Response · Authors · 2025-11-22
> **Response to Reviewer YqAF (Part 2)**
>
> >Q1: I noticed the paper's discussion on the distributional inconsistency  between generated data and real data. Could you elaborate on the data  construction pipeline used during fine-tuning? Specifically, what are  the sources of the images, text, and audio in the fine-tuning dataset?
>
> We only use synthetic data during pretraining, where our goal is to learn strong representation extraction capabilities from our high-quality generated dataset. For all downstream evaluations (audio classification, audio–visual retrieval, localization), we necessarily fine-tune or perform linear probing on real data, since the projection head must be adapted to the distribution of the evaluation sets. No synthetic samples are used during fine-tuning.
>
> > Q2: The current generation models mentioned in the paper, such as FLUX and  Stable Audio Open, also rely on contrastive learning models for prompt  understanding (these models include pre-trained contrastive learning models in their structures). Essentially, can the paper's method be  considered as a distillation and integration of semantic modality alignment within existing generative models? If so, can the paper's  method be compared in terms of model performance with methods like  Ex-MCR that also integrate existing contrastive learning models?
>
> Modern generators like FLUX and Stable Audio Open indeed contain contrastive encoders internally to interpret prompts, but their objective is high-fidelity generation, not AV alignment. Our method differs in both:
>
> - Objective: Existing generators do not explicitly align audio and visual modalities; they generate single-modality content conditioned on text. We use them to repair mismatched AV pairs by injecting language-level semantic alignment.
>
> - Mechanism: We do not distill features or logits from these encoders into downstream discriminative models. Instead, we use the generators strictly as data producers. Alignment is introduced at the dataset level, which can then benefit any AV learner, including those that do not include contrastive modules.
>
> Regarding Ex-MCR, as it directly integrates multiple pretrained contrastive spaces within its training loop, its goal are different from ours. It is mainly focus on building a good retrieval model across modalities. Our contribution is an orthogonal, well-aligned synthetic dataset that improves any downstream task without modifying model architectures. Besides audio–visual retrieval, we can also help with tasks like audio classification, and sound localization.

---

### Author Response · Authors · 2025-12-04
**Summary of Rebuttal**

Dear Area Chair,

Thank you for coordinating the review process. Because the rebuttal phase was interrupted, we provide a brief summary of our responses. We carefully addressed all concerns raised by all four reviewers, adding clarifications, analyses, and additional experiments where needed.

Reviewer YqAF
We clarified our data-quality assessment, highlighting that the paper already includes (1) a human study of semantic alignment and (2) an automatic semantic-similarity evaluation using Sentence-BERT, consistent with the reviewer’s suggestion. We explained why CLIP/ImageBind-based evaluation is biased toward real images. We distinguished our method from retrieval-based approaches such as Ex-MCR: they align existing representations, whereas we generate new cross-modal pairs, and architecture works such as ImageBind and LanguageBind, where our method focus on data and works with all kinds of existing representation learning methods. We clarified that synthetic data is used only for pre-training, and that generators are not used for distillation.

Reviewer ExQR
We justified our choice of baselines for sound localization and provided the exact prompt templates for caption refinement to ensure reproducibility. We reported image generation diversity analysis with Vendi scores and explained our empirical prompt-selection strategy. We added audio-only VGGSound results and discussed how synthetic data repairs failures in rare or mismatched categories.

Reviewer vJDQ
We explained our choice of generators and summarized ablations showing consistent benefits across models. We provided new results using our synthetic AudioSet variant, surpassing real data on both audio classification and audio-visual classification tasks. We included missing citations to recent audio–visual models and clarified the reason to mainly stick to single frame; nonetheless, a multi-frame ablation shows our data also benefits settings with limited temporal context.

Reviewer sxkD
We clarified the novelty: prior work uses language to align modalities in representation space, while we use it to generate corrected cross-modal data, making our contribution a dataset-level solution. We justified our single-frame design, acknowledging video as future work, and showed that synthetic data also helps on fine-grained tasks like localization. We provided guidance on mixing ratios, described our conservative caption-rewriting procedure, and discussed limitations for extremely fine-grained audio tasks.

All clarifications, examples, prompts, and additional experiments have be incorporated in the revised version. We hope this summary helps you to contextualize the reviews.

Sincerely,
The Authors

---

### Meta-Review · Area_Chair_zxkL · 2026-01-07

**Summary:**

In my view, the problem studied in this paper is significant and valuable. Audio-visual learning (and more broadly any cross-modal learning) makes a number of assumptions about the relationship between modalities within and across datapoints, and studying these assumptions is important. This paper proposes a simple and thoughtful pathway to this, and has a reasonable set of initial experiments.

I find it difficult to support accepting the paper based on the reviewers’ analyses, as noted below. While there is not one sweeping concern which invalidates the paper, there are a number of more modestly scoped unanswered questions which make it more challenging to interpret and make use of the results of the paper (data quality assessment, temporal consistency, low-level task trade-offs, noisy generation, synthetic:real ratio, etc.).

**Reviewer Concerns:**

### Meaningfully addressed by the author response:
- **YqAF**: W1 data quality assessment (partially); W2 comparison scope; Q1 pretraining
- **ExQR**: W1 not using SOTA models to compare (I don’t think this is a valid concern in the first place, for the reason the authors noted); W3 sample diversity/noise (partially); W4 VGGSound audio evaluation
- **vJDQ**: W1 frontier generation models; W2 AudioSet; W3 citations (though the reviewer does not clarify the specific relevance of these references so I doubt the validity of this concern)
- **sxkD**: W1 technical novelty; Q2 captioning

### Remaining concerns:
- **YqAF**: W1 data quality assessment (partially): I am not persuaded by the argument re: CLIP/ImageBind’s real-image bias (I believe this is true, but this is [a] not reason enough to skip this evaluation i.e. it is better as a caveat on whatever these numbers show, and [b] confounded with quality and other factors, i.e. we cannot isolate realism enough to causally declare bias).
- **YqAF**: Q2 relationship to contrastively trained encoders in generative models. Though I acknowledge what the authors note about not directly distilling characteristics of the encoders, it is unclear how their features shape the results i.e. we don’t observe the counterfactual of how the generated data might differ without these contrastively trained encoders. However, I also know this is difficult to estimate. It is not a major point against the paper, but I do think the concern itself has not been addressed.
- **ExQR**: W2 prompts and reproducibility. The design process for the prompts, the rationale for the components, etc. are simply not laid out. For example, what does “all possible sound events” mean vs. “all sound events” which seems more accurate? Similar questions can be asked of the second prompt provided. To the degree a paper relies on prompts, it is important these be specified thoroughly and justified to the degree feasible. This isn’t a major concern, but it is one that I think has not been adequately addressed.
- **ExQR**: W3 sample diversity/noise (partially): Though I think the diversity issue is addressed, the empirical approach to mitigating the impact of noise seems like a bandaid and does not address structural causes of noise (e.g. one prompt mapping to multiple possible generated samples). Once again not a major concern, but I don’t think the response addresses this problem directly.
- **vJDQ**: W4 video. The multi-frame ablation is nice to have, but it doesn’t address the core problem of temporal consistency. The author response is vague about this; though I agree that in principle it confounds video generation quality with the goal here, it is difficult to accept the argument in 2025/6 that AV representations are often single-frame (e.g. AudioCLIP is quite old now). I think the response downplays the importance of the video/multi-frame component, even though I do think there are non-trivial challenges in realizing it in practice even as an extension.
- **sxkD**: W2 video, as before
- **sxkD**: W3 low-level tasks. Any trade-offs incurred (as the authors note this approach does specialize in higer-level representations) should be made clear. The sound localization results are encouraging, but this is one task. While not a major concern, it does limit our knowledge of the trade-offs
- **sxkD**: Q1 search for ratio. The range specified by the authors is rather large, which suggests that indeed the best way to find the optimal ratio is by search.
- **sxkD**: Q3 related to W3

**Reviewer Scores:**

Reviewers ExQR and vJDQ at least, in my view, have some cause to raise their scores (but also have cause to maintain them). I suspect the other reviewers would maintain their scores.

---

### Decision · Program_Chairs · 2026-01-26

Reject